# Unveiling Drivers of Retinal Degeneration in RCS Rats: Functional, Morphological, and Molecular Insights

**DOI:** 10.3390/ijms25073749

**Published:** 2024-03-28

**Authors:** Kabir Ahluwalia, Zhaodong Du, Juan Carlos Martinez-Camarillo, Aditya Naik, Biju B. Thomas, Dimitrios Pollalis, Sun Young Lee, Priyal Dave, Eugene Zhou, Zeyang Li, Catherine Chester, Mark S. Humayun, Stan G. Louie

**Affiliations:** 1Mann School of Pharmacy and Pharmaceutical Sciences, University of Southern California, Los Angeles, CA 90089, USA; kahluwal@usc.edu (K.A.); aanaik@usc.edu (A.N.); priyalda@usc.edu (P.D.); eugenezh@usc.edu (E.Z.); zeyangli@usc.edu (Z.L.); chesterc@usc.edu (C.C.); 2USC Ginsburg Institute of for Biomedical Therapeutics, University of Southern California, Los Angeles, CA 90033, USA; duzhaodong@126.com (Z.D.); juan.martinez@med.usc.edu (J.C.M.-C.); biju.thomas@med.usc.edu (B.B.T.); pollalis@usc.edu (D.P.); sunyoung.lee@med.usc.edu (S.Y.L.); humayun@med.usc.edu (M.S.H.); 3Roski Eye Institute, Keck School of Medicine, University of Southern California, Los Angeles, CA 90033, USA; 4Department of Physiology & Neuroscience, Keck School of Medicine, University of Southern California, Los Angeles, CA 90089, USA

**Keywords:** retina, retinal degeneration, oxidative stress, inflammation, citrullination, Royal College of Surgeons rat, retinal imaging, electroretinography, retinal vascular diseases, inherited retinal diseases

## Abstract

Retinal degenerative diseases, including age-related macular degeneration and retinitis pigmentosa, significantly contribute to adult blindness. The Royal College of Surgeons (RCS) rat is a well-established disease model for studying these dystrophies; however, molecular investigations remain limited. We conducted a comprehensive analysis of retinal degeneration in RCS rats, including an immunodeficient RCS (iRCS) sub-strain, using ocular coherence tomography, electroretinography, histology, and molecular dissection using transcriptomics and immunofluorescence. No significant differences in retinal degeneration progression were observed between the iRCS and immunocompetent RCS rats, suggesting a minimal role of adaptive immune responses in disease. Transcriptomic alterations were primarily in inflammatory signaling pathways, characterized by the strong upregulation of *Tnfa*, an inflammatory signaling molecule, and *Nox1*, a contributor to reactive oxygen species (ROS) generation. Additionally, a notable decrease in *Alox15* expression was observed, pointing to a possible reduction in anti-inflammatory and pro-resolving lipid mediators. These findings were corroborated by immunostaining, which demonstrated increased photoreceptor lipid peroxidation (4HNE) and photoreceptor citrullination (CitH3) during retinal degeneration. Our work enhances the understanding of molecular changes associated with retinal degeneration in RCS rats and offers potential therapeutic targets within inflammatory and oxidative stress pathways for confirmatory research and development.

## 1. Introduction

Animal models of retinal degeneration are indispensable tools for investigating the underlying molecular substrates of dystrophies, such as age-related macular degeneration (AMD) and retinitis pigmentosa (RP). Additionally, correlating the manifestations of the disease in animal models and humans is critical for the development of therapeutics. In this context, the dystrophic Royal College of Surgeons (RCS) rat is one of the most utilized models for evaluating the efficacy of new therapies to treat retinal dystrophies such as AMD and RP [1,2,3,4]. The RCS rat has a *Mertk* (MER Proto-Oncogene, Tyrosine Kinase) gene mutation, leading to retinal pigment epithelium (RPE) cell phagocytic dysfunction, which ultimately leads to photoreceptor degeneration. Compared with rhodopsin mutation rat models, such as P23H and S334ter rats, the RCS rat model represents a more generalized model of retinal degeneration due to RPE dysfunction, which is a core characteristic of AMD and RP [5].

In recent years, RCS rats have gained prominence in the investigation of stem cell transplantation for AMD, partly because of their size advantage over murine eyes, allowing for easier surgical procedures, and the availability of immunocompromised strains such as the *Foxn1^mu^* immunocompromised RCS rat (iRCS) [6,7,8,9,10,11]. Molecular analysis of disease models is a springboard for the development of new therapeutic strategies. However, previous observations in RCS rats have mostly focused on anatomical and functional changes, such as photoreceptor death associated with the *Mertk* mutation in RPE [12,13]. It should be noted that microglia also express *Mertk*, but the defect in microglial phagocytosis and its contribution to retinal degeneration are unclear [12]. Therefore, this study aimed to investigate the molecular changes in the neural retina secondary to the primary *Mertk* mutation and how they are altered during the progression of retinal degeneration in the RCS disease model. Our objective was to dissect the molecular mechanisms of the neural retina involved in retinal degeneration, which will contribute to the development and evaluation of novel treatment strategies. Here, we present an examination of the existing detection methods (ocular coherence tomography (OCT), electroretinography (ERG), and histology) and molecular characterization (mRNA transcriptomics, immunodetection, and RT-qPCR) throughout the course of retinal degeneration in RCS and iRCS rats. Our goal was to build a baseline of molecular changes in RCS and iRCS rats to promote further research and therapeutic development for specific molecular targets to treat retinal degeneration. 

## 2. Results

### 2.1. Structure and Function Loss with Age in RCS and iRCS Rats

The longitudinal retinal morphology of the RCS and iRCS rats was evaluated using H&E-stained retinas, photoreceptor counts, and OCT at postnatal day 21 (p21), p35, p49, p60, and p90 (RCS in Figure 1; RCS and iRCS in Appendix A). The thicknesses of all retinal layers decreased across the evaluation time points. In particular, the outer retina and ONL showed significant gradual thinning at all the time points. Similarly, OCT imaging was able to detect significant gradual thinning of all retinal layers at each time point (Figure 1G–J). However, OCT analysis revealed initial thickening of the outer retina at p35. By p60, the ONL had degenerated and consisted of four rows of nuclei. Additionally, we compared the immunocompetent RCS and immunocompromised RCS (iRCS) sub-strains, and no statistical differences were noted between the two sub-strains (Appendix A). However, our study may be insufficiently powered to detect minor structural and functional differences between RCS and iRCS.

Retinal function was evaluated using scotopic and photopic ERG. Representative ERGs at 3.0 cd∙s/m^2^ for the RCS are summarized in Figure 2; There’s no difference between RCS and iRCS in ERG readings (Appendix A). All waveforms were measurable, except for the photopic a-wave, which had amplitudes of less than 5 µV at all time points. The scotopic a-wave dropped rapidly in the early stages, with a 72% decrease in amplitude from p21 to p35 and an additional 20% decrease from p35 to p49, where it reached its minimum (Figure 2A). In contrast, the scotopic b-wave had a slower decline, with a ~43% decrease between p21 and p35, 33% between p35 and p49, and a non-significant 20% decline from p49 to p60 (Figure 2B). The decline in photopic b-wave was only 8% between p21 and p35, but it dramatically declined by approximately 56% and 24% from p35 to p49 and p49 to p60, respectively (Figure 2E). Representative scotopic and photopic ERG waves are shown in Figure 2C and Figure 2F, respectively. As photopic ERG is a measurement primarily of the cones and downstream retinal neurons in the cone signaling pathway, the delayed photopic b-wave decline is consistent with literature showing that rods degenerate first in RP [14]. Overall, histology, OCT, and ERG showed a strong association with each other in relation to the aging of RCS rats.

### 2.2. Identification of Differentially Expressed Genes (DEGs) and Cell Types

RNA-Seq of whole-retina samples from RCS and iRCS rats was performed at p21, prior to retinal degeneration, and at p60, where structural photoreceptor degeneration was evident (*n* = 3 at each time point for each group). Differentially expressed genes (DEGs) were analyzed using an absolute fold change cutoff of 2 and an adjusted *p*-value cutoff of 0.05 (fold change > 2, p_adj_ < 0.05). Several differences in the transcriptome between RCS and iRCS rats were observed at p21 or p60 (Figure 3A,B); however, these were minimal relative to the analysis between p21 and p60 in the RCS rats (Figure 3C) and iRCS rats (Figure 3D). Between p21 and p60, there were 1009 and 839 significantly downregulated (Figure 3C,D) and 909 and 869 significantly upregulated (Figure 3C,D) DEGs for RCS and iRCS, respectively. This further supports the lack of significant differences between the sub-strains, which was also demonstrated in principal component analysis (PCA), sample distance, and biclustering plots (Appendix A). 

We hypothesized that photoreceptor cells, which decrease in number throughout disease progression, are the major contributors to the significantly downregulated DEGs and confounded the pathway analysis. To assess this, genes were labeled with a retinal cell type based on previously published single-cell RNA-seq data, as described in the Materials and Methods Section 4.5. Overall, 6666 genes were labeled with a specific cell type (summarized in Table 1 for RCS and Table 2 for iRCS). As expected, when the data were filtered for significant DEGs, the largest proportion of downregulated genes (57% for RCS and 60% for iRCS) belonged to photoreceptor, rod, and cone cell types. The fact that downregulated genes are primarily associated with photoreceptors is confounded by the fact that this is a photoreceptor degeneration model; as such, no inferences can be made on the transcriptomic changes in photoreceptors using this bulk-sequenced RNA. Microglia, macroglia (Müller glia and astrocytes), and vascular cells contributed to a high proportion of upregulated DEGs, with 31%, 37%, and 11% for RCS and 45%, 28%, and 12% for iRCS, respectively. While we may have correctly assigned DEGs to specific cell types, the assignment of genes to specific cell types using bulk-sequenced retinas is merely an exercise in understanding our transcriptomic data when a large proportion is confounded by the degeneration of photoreceptors. Ultimately, these data require single-cell sequencing for confirmation. For example, DEGs assigned to microglia have a large crossover with infiltrating immune cells, which may be distinguishable using single-cell analysis. The full datasets and the cell-type filtered data were analyzed for pathway enrichment using QIAGEN Ingenuity Pathway Analysis (IPA). 

### 2.3. Pathway Analysis of RCS Retinal Degeneration

Significant DEGs were analyzed for pathway enrichment using IPA. Unsurprisingly, visual phototransduction was the most significantly altered pathway for both RCS and iRCS (Figure 4 and Appendix A, respectively). Other top enriched pathways for both RCS and iRCS rats included phagosome formation, G-protein coupled receptor signaling, neuroinflammation, triggering receptor expressed on myeloid cell 1 (TREM1) signaling, and cytokine storm signaling. Most of the top enriched pathways were inflammatory response pathways and were predicted to be activated. In addition, the anti-inflammatory IL-10 pathway was enriched and predicted to be inhibited in both RCS and iRCS rats (all the enriched pathways are listed in Appendix A). 

We focused on the subset of DEGs labeled with microglial cell types due to the percent change from all genes (12%) to upregulated DEGs (31% for RCS and 45% for iRCS). We performed pathway analysis for the subset of DEGs labeled with microglial cell type for both RCS and iRCS (Figure 5 and Appendix A, respectively). The top two enriched pathways were neutrophil degranulation and phagosome formation in both the RCS and iRCS rats. Similar to the total dataset, most pathways enriched in microglia were activated pro-inflammatory pathways. In addition to the neutrophil degranulation pathway, the neutrophil extracellular trap (NET) pathway was enriched in both RCS and iRCS. Other pathways, such as phagosome formation, TREM1 signaling, toll-like receptor cascades, and other inflammatory pathways are known to be linked to NETosis [15,16,17]. There may be similarities between NETs and macrophage extracellular trap pathways that explain this enrichment [18]. However, this analysis utilizes bulk-sequenced tissues, and is a temporary analysis to be confirmed by further single-cell analysis.

These pathways involve inflammatory signaling through TNF receptors, TLRs, Fc receptors, complement receptors, non-complement-receptor integrins, lectins, and immunoglobulins. Consistent with pathway enrichment, the top genes upregulated in microglial cells included lectins (Clec5a and Clec7a), integrins (Icam1 and Itgam), MHCII (Hla-dra), and CSF receptor (Table 3 for RCS and Table 4 for iRCS, full dataset in Appendix A). Several TLRs were also upregulated in this dataset, as confirmed by RT-qPCR and densitometry (Appendix A). Interestingly, one of the most significantly downregulated genes in the microglial dataset was Alox15, which encodes the 15-lipoxygenase (15-LOX) enzyme involved in polyunsaturated fatty acid (PUFA) metabolism. The 15-LOX enzyme is critical for the formation of SPMs that control inflammatory resolution, suggesting that the loss of 15-LOX could contribute to chronic and uncontrolled inflammation associated with retinal degeneration.

### 2.4. Inflammation and Oxidative Stress Increase with Retinal Degeneration

Changes in the RCS transcriptome suggest that oxidative stress and inflammation are key components in retinal degeneration; however, the changes in gene expression associated with these mechanisms in a longitudinal study of RCS rats have not been previously evaluated. Here, we compared retinal inflammation gene expression (*Tnfa* and *Nfkb1*), reactive gliosis genes (*Gfap*), ROS-generating genes from the NADPH oxidase (NOX) family (*Nox1*, *Nox2*, *Nox4*, and *Cyba*), and antioxidant genes (*Sod1*, *Sod2*, *Sod3*, and *Cat*) throughout retinal degeneration in the RCS and iRCS models (Figure 6; scatter plots with statistics are shown in Appendix A). 

Both RCS and iRCS showed a similar pattern for these genes, with *Tnfa* and *Nox1* being the two most dramatically upregulated genes in both the datasets. Both genes were upregulated by approximately 20-fold by p35. At p49, *Tnfa* increases to 34-fold and 44-fold compared to p21 for RCS and iRCS, respectively, while *Nox1* increases to 67-fold and 80-fold compared to p21 for RCS and iRCS, respectively. *Gfap* showed delayed increased expression until p49 but continued to increase at p60. *Tgfb1*, which can exert anti-inflammatory and pro-inflammatory effects, peaked at p49 and was subsequently reduced by p60. Upregulation of *Tnfa* and *Tgfb1* corresponded with increased expression of ROS-generating genes, such as *Nox1*, *Nox2*, and *Cyba*. The *Nox4* isoform was expressed at low levels and was not upregulated, suggesting that it might not participate in the overall pathology.

Cellular adaptive, protective genes were also evaluated across the time points, where increases in *Sod1* and *Sod2* isoforms were increased by 1.5-to 2.4-fold for RCS and 1.3-to 1.5-fold for iRCS at the same time points (p35). However, these changes in gene expression may not be sufficient to manage the oxidative stress. In addition, the extracellular *Sod3* isoform did not show any upregulation at any time point except for one animal at p60 for both RCS and iRCS, suggesting that inadequate handling of extracellular oxidative stress may be driving retinal disease progression. This pattern of expression indicates a shift to a pro-inflammatory and pro-oxidative microenvironment in the retina during degeneration.

### 2.5. Oxidative Products Increase during Retinal Degeneration

To further assess the oxidative damage associated with NOX gene expression, retinal lipid peroxides, malondialdehyde (MDA), and 4-hydroxynonenal (4HNE) were investigated (Figure 7). These lipid peroxides are reactive metabolites of PUFA oxidation, and we observed increased staining of 4HNE, but not MDA, in the ONL of RCS and iRCS rats. However, at the earliest time point, strong staining was observed in the GCL and choroid, suggesting early oxidative stress in the retinal vasculature (Figure 7A). Additionally, the inner segments of the photoreceptor stain for 4HNE at p21, and as degeneration of the inner segments occurs, this stain is gradually lost. 

MDA primarily appeared in the nucleus (Figure 7A, white arrows). In contrast, 4HNE is primarily cytoplasmic or extracellular, and does not co-label with DAPI. Additionally, the 4HNE intensity in the ONL increased at each time point. Regions of the external limiting membrane (ELM) at p60 were noted to have increased staining of MDA and 4HNE which spilled over into the inner segments and outer segments (IS/OS, Figure 7A white triangles). The upregulation of 4HNE at p35 corresponds to increased expression of ROS-generating genes, such as *Nox1*, *Nox2*, and *Cyba*. In addition, MDA and 4HNE can act as DAMPs that activate pro-inflammatory signaling through TLR2/4, which is part of the TREM1 signaling pathway [19]. Both mRNA and protein expression of TLR4 were measured and showed increased expression with age for RCS and iRCS; however, iRCS only showed a non-significant trend upward for protein expression. Additionally, RNA-sensing TLR7 and DNA-sensing TLR9 were upregulated with age. TLR9 showed significantly increased mRNA expression and reached the highest increased protein expression of 15-fold at p60 compared to p21 (Appendix A).

### 2.6. Retinal Citrullination Is Associated with Degeneration

Recently, peptidyl arginase deiminase 4 (PAD4) was identified in reactive gliosis during retinal degeneration [20]. Our RNA-Seq identified neutrophil degranulation and NETosis as enriched pathways in the microglial dataset. A major contributor to neutrophil degranulation is PAD4 and is a nuclear-targeting enzyme that citrullinates DNA histones. In addition, DAMP-mediated TLR signaling results in NOX activation and ROS generation, leading to PAD4 activation and nuclear citrullination [21,22]. We evaluated the expression of PAD4 and its product, citrullinated histone h3 (CitH3), in RCS and iRCS retinas throughout retinal degeneration (Figure 8 and Figure 9). At p21, PAD4 is confined to the GCL and INL. PAD4 began to expand into the outer retina, with increased OPL staining by p35 (Figure 8A). Strong staining was observed throughout the ONL by p49, with intense staining of the ELM and faint staining below the ELM. Additionally, choroidal staining of PAD4 was observed at p49, suggesting both intra- and extra-retinal sources of PAD4. At p60, ONL staining was similar, but there were regions of the ELM where PAD4 extended into the IS/OS, consistent with recent findings that PAD4 is related to Müller cells and reactive gliosis (Figure 8 white triangles) [20]. ImageJ version 1.53a quantification confirmed these observations, showing a significant increase in ONL at p49 and p60 in RCS rats (Figure 8B). However, iRCS did not show a significant increase in ONL staining (Figure 8C).

CitH3 is a byproduct of PAD4 activity and was analyzed in the RCS and iRCS retinas (Figure 9). At p21, minimal detection of CitH3 was observed in the photoreceptor nuclei. By p35, there was a significant upregulation of CitH3 staining of photoreceptor nuclei, and the intensity of the stain increased further at p49 (Figure 9A). For RCS rats, there was a statistically significant increase in ONL staining of CitH3 at p49, which peaked with a mean increase of 148-fold compared to p21 before returning to low staining at p60 (Figure 9B). For iRCS, only one out of three animals stained strongly at p49 (Figure 9C). As PAD4 was observed at the earliest time point, it is likely that early activation of basal levels of PAD4 between p21 and p35 results in CitH3 staining before the significant translocation of PAD4 into the ONL. Additionally, there was heterogeneity in the staining intensity of the individual photoreceptor nuclei at p35 and p49. The intensity of the staining may be an indication of cells undergoing cell death, which showed more intense staining at p49. Interestingly, photoreceptor nuclei at p60 had limited staining, but at both p49 and p60, CitH3 staining was observed in IS/OS. These findings were confirmed by western blot analysis of retinal tissues, in which CitH3 was not observed at p21 but was clearly detected at p35–p60 (Figure 9D,E and Appendix A). As opposed to the peak in staining at p49, western blots suggested a continued accumulation of CitH3. Because there is no RPE phagocytosis in the RCS rats, extracellular citrullinated DNA may continue to accumulate in the IS/OS. Overall, these results suggest a possible involvement of PAD4 in photoreceptor degeneration.

## 3. Discussion

Retinal degenerative diseases, including AMD and RP, are leading causes of adult blindness worldwide [23]. RCS rats have been widely utilized as animal models for retinal degeneration and have been extensively characterized using histology, OCT, and ERG [24]. However, longitudinal studies evaluating the molecular mechanisms underlying retinal degeneration in RCS models are lacking. Our study offers preliminary insights into the molecular events in the RCS retina, suggesting potential targets for future therapeutic development. In addition to structural and functional assessments, this study evaluated the transcriptomics of the retinal tissue before and after retinal degeneration with supplemental RT-qPCR and immunochemistry. 

Although the systemic immune system plays a role in certain retinal pathologies [25], T-cell deficiency in the iRCS model does not alter morphological or functional retinal degeneration characteristics. This is also mirrored by our transcriptomic analysis, which showed minimal differences between the RCS and iRCS rats, and strong similarities when comparing enriched pathways and top DEGs. As expected, our results confirmed a reduction in retinal thickness measured by OCT in both the inner and outer retinal layers, as supported by histological findings, and declines in ERG waveforms, including scotopic and photopic a-wave and b-wave amplitudes [26,27]. Overall, histology, OCT, and ERG findings are consistent with the literature, providing valuable data to guide future interventional studies. 

Following the characterization of functional and structural changes associated with retinal degeneration, we aimed to investigate longitudinal changes in molecular markers of the neural retina, beginning with an agnostic transcriptomic approach. Because the analysis would be confounded by the loss of photoreceptors, we utilized single-cell transcript assignments from previous studies to separate transcripts based on cell types. As expected, photoreceptors were the majority of downregulated DEGs, and no pathway analysis was performed on photoreceptors. Interestingly, a large percentage of the upregulated DEGs (31% for RCS and 45% for iRCS) were associated with microglia and prompted us to focus on microglia for transcriptomic analysis. Pathway enrichment analysis of the total dataset compared to cell type-specific analysis revealed that microglia primarily contribute to the pathogenic transcriptome changes associated with RCS retinal degeneration, which were enriched for inflammatory signaling pathways. This finding is consistent with a recent single-cell RNA-Seq analysis of human AMD conducted by Menon et al., who identified cone photoreceptors, macroglia, microglia, and vascular cells as the most predictive factors for AMD risk [28]. The focus on microglia is further supported by studies which have shown that microglia are activated early in retinal degeneration, accelerate the loss of photoreceptors, and that inhibiting microglia activation reduced pro-inflammatory cytokine expression and preserved photoreceptors [14]. We have expanded on this knowledge by identifying specific pathways that can be targeted as described below.

Only one previous transcriptomic analysis of RCS has been conducted, comparing RCS to Long Evans rats [29]. Our study identified comparable activated pathways, with an emphasis on microglial cell activation; however, Jones et al. [29] only detected downregulation of visual function pathways. In contrast, our study found downregulation of IL10 and peroxisome proliferator-activated receptor (PPAR) signaling, and coordinated lysosomal expression and regulation (CLEAR) signaling (Appendix A). PPAR and CLEAR signaling are regulators of inflammation, where PPAR agonism by PUFAs results in anti-inflammatory effects, and CLEAR controls lysosomal biogenesis, autophagy, and cellular clearance [30,31,32]. Overall, this indicates a decrease in anti-inflammatory pathways and cellular debris clearance. Our results suggest that microglia may be pathogenic contributors to transcriptomic changes that occur during retinal degeneration. As mentioned previously, the assignment of cell types to bulk-sequenced tissues is a stopgap and can be confirmed by single-cell profiling. This is primarily important in disease models, as cells may activate the expression of genes that are atypical for that cell type and the proportions in cell types are dynamic. Future studies utilizing single-cell RNA-Seq can further improve the transcriptomic profiling of microglia in RCS rats. 

This study suggests that inflammation, gliosis, and oxidative stress are major contributors to retinal degeneration [33]. ROS-generating gene expression peaked and plateaued at p49, coinciding with the minimum scotopic a-wave amplitude. Among the genes that showed the highest upregulation with RCS age were *Gfap*, *Tnfa*, and *Nox1*. *Gfap*, while not a biomarker for inflammation or oxidative stress, is a common marker for retinal damage and is expressed by macroglia. Upregulated *Gfap* expression is observed when Müller glia fill retinal breaks during the formation of gliotic scars and can be observed in IS/OS following photoreceptor cell death [34]. Our findings indicate that *Gfap* expression is not upregulated until p49 when significant photoreceptor loss has already occurred. While reactive gliosis has been suggested as a pharmacological target for inhibition [35], in the RCS model gliosis might be a secondary condition and inhibiting it would only slow down late-stage neuronal degeneration. *Tnfa*, a proinflammatory cytokine expressed by immune cells like microglia and monocytes in the retina [36], showed high expression at p35, indicating a potential early response of microglia to retinal stress. *Nox1*, an isoform of the gp91phox catalytic subunit of NOX, was significantly upregulated at p35 and peaked at p49. While other isoforms, such as *Nox2* and *Nox4*, are considered relevant to other vascular pathologies [37], *Nox1* plays a central role in degeneration in the RCS model, with minor contributions from *Nox2* and the *Cyba* subunits. Studies in a mouse model have demonstrated that *Nox1* knockout, but not *Nox2* or *Nox4* knockout, protects the retina against oxygen-induced retinopathy and identifies microglia as the source of hypoxia-induced ROS generation and neovascularization [38]. These findings hint at the microglial response potentially playing a role in the early phases of retinal degeneration through inflammatory signaling and ROS generation, warranting further study.

In contrast, the SOD-family genes showed only mild upregulation, indicating that protective antioxidant mechanisms are insufficient to manage the accumulation of ROS. Unmanaged ROS can react with cellular components to form toxic metabolites, such as MDA and 4HNE, which can propagate cellular damage by forming adducts with DNA or proteins and initiate inflammatory signaling [39]. 4HNE demonstrated a time-dependent increase in ONL staining, with a significant increase at p35, similar to *Nox1* gene expression. Our data suggest that the ONL is the primary site of increased oxidative stress compared to the other retinal layers. 

MDA and 4HNE are derived from PUFA oxidation, such as DHA, which is present in high concentrations in the retina, and decreased retinal DHA is a risk factor for retinopathies [40]. Furthermore, MDA can form adducts with glial proteins such as *Gfap*, vimentin, and glutamine synthetase [41], and we observed strong MDA staining in the GCL, where the endfeet of Müller glia are localized. Additionally, 4HNE has been linked to protein modifications in neurodegenerative diseases, contributing to altered energy metabolism, mitochondrial dysfunction, and insufficient antioxidant mechanisms. It is also known to induce the synthesis of TNFα and TGFβ [42,43]. There is an age-dependent increase in TLR4 in the RCS retina, which produces ROS and inflammatory cytokines upon receptor binding to DAMPs, including MDA and 4HNE adducts [44]. This information provides new insights into the timing of molecular changes in the RCS model, highlighting microglia as early contributors of pathological gene expression, likely driven by DAMP signaling. Although inhibition of DAMP signaling may not ameliorate the underlying RPE defect, a switch from pro-inflammatory to pro-resolution microglial activity could significantly affect disease progression. We recently confirmed that there is a reduction in MDA and 4HNE staining corresponding to preservation of retinal structure and function in the iRCS rat [45]. 

Neutrophil degranulation and NETosis were highly enriched in the microglia dataset. Increased gene and protein expression of TLR7 and TLR9, which is activated by RNA and DNA debris, respectively, was also observed. TLR signaling in immune cells leads to ROS production, including oxidized lipids, and subsequent ET formation through PAD4. As well, PAD4 and ET have been implicated in retinal degeneration [22,46,47]. Investigation of PAD4 in the retina has focused on citrullination of *Gfap* and the process of gliosis, but PAD4 is also known to translocate to the nucleus and citrullinate histones to induce DNA unwinding expulsion in ET formation [48]. We investigated the changes in retinal PAD4 and CitH3 levels throughout retinal degeneration. PAD4 staining was strong in the INL at p21, and it gradually translocated toward the ONL, exhibiting strong staining in the ONL at p49 and p60. Interestingly, photoreceptor CitH3 significantly increased at p35 before significant ONL PAD4 staining. This may be due to increased activation of basal levels of calcium-dependent PAD4 before significant upregulation. TNFα, NOX, and mitochondria-produced ROS are known initiators of PAD4-mediated DNA release [49], both of which were upregulated at p35 similar to CitH3. We recently showed citrullination is inversely correlated to retinal preservation in the iRCS rat [45], suggesting a novel treatment mechanism for retinal degeneration. Additionally, we recently reported that the administration of a specialized pro-resolving lipid mediator (SPM), a lipoxin A4 analog, upregulated *Alox15,* one of the top downregulated DEGs in microglia, and significantly increased PUFAs DHA and DPA in RCS rats [50]. We previously reported that this lipoxin A4 analog also reduced neutrophils and NETosis in colorectal cancer mouse models, including the reduction in intratumoral PAD4 and CitH3 [51]. Together, these publications suggest that the pathways and genes identified in our study are targets for further therapeutic development. 

Our study had several limitations. We chose to exclude the RPE in our current study due to the breadth and depth of the literature on RPE [5,52]; however, because of the intimate relationship between the neural retina and the RPE, there is a gap in our analysis with regard to RPE molecular changes. While we found significant results for many of our analyses, our data may not be sufficiently powered to identify differences between the RCS sub-strains. Given the limited size of our dataset, our capacity to correlate molecular changes with structural and functional alterations in the retina is constrained, including the risk of overfitting and diminished power to detect true associations due to necessary corrections for multiple testing. We recognize these limitations and are planning further research to expand upon our preliminary findings, aiming to incorporate larger datasets that may allow for more extensive correlation analyses and a deeper understanding of retinal degeneration mechanisms.

Although the RCS rat model is widely used and provides valuable insights into retinal degeneration, it is essential to acknowledge that animal models may not fully recapitulate the complexity of human retinal diseases. Differences in genetic background, physiological factors, and disease progression patterns can influence the translatability of findings to human conditions. Therefore, caution should be exercised when extrapolating these results to human retinal pathologies. Clinical data from patients with retinal degenerative diseases are necessary to confirm the relevance and significance of the identified molecular pathways and potential therapeutic targets. Although our study aimed to assess longitudinal changes in molecular markers, it is important to note that the specific time points and intervals chosen may not capture the entire spectrum of retinal degeneration. Disease progression can exhibit dynamic and non-linear patterns, and additional time points, particularly earlier time points, could provide a more comprehensive understanding of the molecular changes over time. Although our study investigated cell-type-specific gene expression patterns, the assignment of cell types based on previous studies introduces an inherent level of uncertainty. Single-cell transcriptomics is a rapidly evolving field, and advancements in technology and techniques may further refine the classification and identification of cell types. Future studies employing more refined methodologies, such as single-nucleus RNA sequencing, may provide deeper insights into the specific cell populations and their contributions to retinal degeneration [53].

In summary, our systematic evaluation of retinal functional and morphological characteristics in RCS rats has shed light on the intricate dynamics of retinal degeneration, with scotopic ERG emerging as a crucial early indicator. While our data underscore the pivotal role of microglial cells and inflammatory pathways in retinal pathology, it is important to interpret these findings within the context of the study’s limitations. Future investigations are essential to validate these insights and should aim to explore specific pathways such as TNFα, NOX1, and citrullination more deeply, using larger and more diverse datasets and advanced analytical techniques. Our complementary publications suggest that strategies aimed at restoring the retinal microenvironment and resolving inflammation hold significant therapeutic promise. However, the translation of these molecular insights into clinical applications requires cautious optimism and further validation. Our study contributes to the understanding of retinal degeneration and sets the stage for future research that could pave the way for innovative therapeutic approaches in retinal and neurodegenerative diseases.

## 4. Materials and Methods

### 4.1. Animals

Animal experiments were conducted in full accordance with the University of Southern California (USC) Institutional Animal Care and Use Committee (IACUC)-approved protocols, the National Institutes of Health Guide for the Care and Use of Laboratory Animals, and the ARVO Statement for the Use of Animals in Ophthalmic and Vision Research. Dystrophic RCS rats were obtained from Dr. Matthew LaVail (University of California, San Francisco, USA) and were bred and propagated at the USC under an IACUC-approved protocol. The iRCS rat breeding pairs were obtained from Dr. Biju B. Thomas which were bred and propagated at the USC under an IACUC approved protocol [2]. All the pups used for this study were their offspring born in the USC vivarium. Rats were group housed under specific pathogen-free conditions and had access to water and food ad libitum. All animals were housed in temperature- and light-controlled rooms with a 12 h light/dark cycle. Experiments were conducted between post-natal day (p) 21 and p90 at which the animal is mostly blind. Additionally, studies were conducted on both pigmented, immunocompetent RCS (RCS-p^+^/RCS-p^+^) and pigmented, immunodeficient RCS (RCS-p^+^/RCS-p^+^ and Foxn1^mu^/Foxn1^mu^) rats to understand the impact immunodeficiency had on disease progression.

### 4.2. Electroretinogram (ERG) Evaluation

Full-field ERG was evaluated using the HMsERG system (OcuScience, Las Vegas, NV) as previously described at the following time points: p21, p35, p49, p60, and p90 [2]. Prior to each evaluation, animals were dark-adapted overnight for 12 h and prepared for testing under dim red light. Rats were anesthetized with a mixture of ketamine 80 mg/kg and xylazine 7.5 mg/kg given as an intraperitoneal injection. Pupils were dilated by topical instillation of 1% tropicamide (Bausch & Lomb Inc., Tampa, FL, USA) and 2.5% phenylephrine hydrochloride (Akorn Pharmaceuticals, Lake Forest, IL, USA). ERG was recorded from both eyes using ERG-Jet contact lens electrodes (Fabrinal SA, La Chaux-de-Fonds, Switzerland). Reference and ground electrodes were inserted into the infraorbital (malar) area and between the ears, respectively. The conductivity between the cornea and recording electrodes was maintained by an optically clear ophthalmic gel (GenTeal Gel, NOVARTIS, containing hydroxypropyl methylcellulose). Scotopic testing was conducted with flash stimuli intensities ranging from 1 to 25,000 millicandela (mcd) followed by photopic testing (flash stimuli responses of 10–25,000 mcd). A 10 min light adaptation period was performed prior. Scotopic and photopic a-wave and b-wave amplitudes were analyzed.

### 4.3. Ocular Coherence Tomography (OCT) Evaluation

After ERG functional testing, spectral-domain OCT (SD-OCT) images were obtained by the diagnostic imaging platform (Spectralis HRA+OCT, Heidelberg Engineering Inc., Heidelberg, Germany). With the animals under anesthesia and their pupils fully dilated, multiple horizontal linear scans were obtained at the central, nasal, and temporal retina. Total retina, inner retina, and outer retina (including an intraretinal segmentation of the outer nuclear layer (ONL)) thickness were measured at five points along the same horizontal line at nasal and temporal regions from the optic nerve head (ONH). Measurements were averaged prior to statistical analysis. The cornea was moist by a frequent application of balanced salt solution (BSS) (Alcon Laboratories, Inc. Fort Worth, TX, USA) during the entire imaging session.

### 4.4. Euthanasia and Tissue Collection

At the end of study, rats were euthanized by intraperitoneal injection of 0.5 mL pentobarbital sodium 390 mg and phenytoin sodium 50 mg (Euthasol; Virbac AH, Inc., Fort Worth, TX, USA). For histology analysis, eyes were enucleated and fixed in Davidson’s solution. After 24 h of fixation, the Davidson’s solution was replaced with 70% ethanol and sent to the USC Ginsburg Institute for Biomedical Therapeutics Core for paraffin embedding, sectioning, and H&E staining. Anterior segment structures, including the cornea, iris, and lens were removed and the posterior pole was exposed. The cut of the eye was through the optic nerve on its sagittal plane. After dissection, all eyes were embedded in paraffin and cut in a microtome starting from the center of the optic nerve. Serial sections of 5 µm in thickness were performed throughout the entire eyeball. For RT-qPCR and Western Blot analysis, retinas were isolated from freshly enucleated eyes and placed into 1 mL RNA-zol RT (Sigma Aldrich, St. Louie, MO, USA) or 0.5 mL T-PER Tissue Protein Extraction Reagent (Thermo Fisher Scientific, Waltham, MA, USA), respectively, and processed as described below. 

### 4.5. Retinal Transcriptomics

Total RNA was isolated, as described below in 4.6 Retinal RT-qPCR, and sent to Azenta Life Sciences (South Plainfield, NJ, USA) for transcriptomics analysis. Sample quality control was performed using a Nanodrop 2000 (Thermo Fisher), Qubit (Invitrogen, Waltham, MA, USA), and TapeStation (Agilent, Santa Clara, CA, USA) and all samples sequenced had an RNA integrity number (RIN) over 8.0. Total RNA underwent polyA selection followed by mRNA sequencing analysis using a HiSeq Illumina-based system with 20–30 million 150 bp paired end reads per sample. RNA-seq data analysis was performed by Azenta including read trimming, mapping, and differential gene expression analysis. Raw data quality was evaluated with FastQC. Reads were trimmed with Trimmomatic and mapped to the reference genome (https://uswest.ensembl.org/Rattus_norvegicus/Info/Index (accessed on 8 August 2022)) with STAR [54]. Gene hits counts were calculated with FeatureCounts and normalized and compared between groups using DESeq2. Mapped genes were labeled with cell types using multiple publications (Appendix A) [28,55,56]. If publications conflicted in cell type assignment, the majority assignment was used or no assignment was made if there was a tie in cell type. Log2 fold change and Benjamini-Hochberg adjusted *p*-values for DEGs were analyzed with the use of QIAGEN IPA (QIAGEN Inc., Germantown, MD, USA, https://digitalinsights.qiagen.com/IPA (accessed on 8 February 2023)) [57].

### 4.6. Retinal RT-qPCR

Isolated Retinas were homogenized in RNAzol^®^ RT (Sigma-Aldrich, St. Louis, MO, USA) using a TissueLyser II (Qiagen LLC, Germantown, MD, USA). Total RNA was extracted following the manufacturer’s instructions and concentration was determined via the NanoDrop™ spectrophotometer (Thermo Fisher Scientific, Waltham, MA, USA). cDNA was prepared using the RevertAid™ First Strand cDNA Synthesis Kit (Thermo Fisher Scientific, Waltham, MA, USA) following the manufacturer’s protocol. The RT-qPCR master mix was prepared by mixing PowerUp™ SYBR™ Green Master Mix (Applied Biosystems, Foster City, CA, USA) and the forward and reverse primers. Primers were designed using Primer-Blast from NCBI and sequences are listed in Appendix A [58]. Diluted cDNA and master mix were pipetted into a 384-well plate using an Assist Plus Pipetting Robot (INTEGRA Biosciences Corp., Hudson, NH, USA) in triplicate. RT-qPCR was performed on QuantStudio 12K Flex Real-Time PCR System (Applied iosystems, Foster City, CA, USA) with the following run method: UDG activation at 50 °C for 2 min, followed by Dual-Lock DNA polymerase at 95 °C for 2 min, then 40 cycles of denature at 95 °C for 15 s and anneal/extend at 60 °C for 60 s, the final stage was a dissociation curve consisting of ramping at 1.6 °C/s to 95 °C for 15 s, then 1.6 °C/s to 60 °C for 1 min and 0.15 °C/s to 95 °C for 15 s. Data was collected and analyzed using the 2^−ΔΔCT^ method using GAPDH as the reference gene [59].

### 4.7. Retinal Western Blot Analysis

Isolated retinas were placed into 500 µL T-PER™ (Thermo Fisher Scientific, Waltham, MA, USA) supplemented with protease and phosphatase inhibitors (Pierce Biotechnology, Rockford, IL, USA) and homogenized using a TissueLyser II (Qiagen LLC, Germantown, MD, USA). Tissue lysates were prepared, and protein concentration was determined using Quick Start™ Bradford protein assay with bovine serum albumin standards (Bio-Rad, Hercules, CA, USA).

For western blots, 10 μg protein was resolved by sodium dodecyl sulfate-polyacrylamide gel electrophoresis using a 4–20% Mini-PROTEAN^®^ TGX™ gel (Bio-Rad, Hercules, CA, USA) with a 10–250 kDa protein ladder (Bio-rad). Protein was then transferred to polyvinylidene fluoride (PVDF) membranes, blocked with 5% nonfat milk, and probed with primary antibodies overnight at 4 °C. A list of antibodies can be found in Appendix A. Following primary incubation, HRP-conjugated secondaries (1:10,000 in 5% nonfat milk) were applied for 1 h at room temperature before incubating in Clarity ECL substrate (Bio-Rad, Hercules, CA, USA) for 5 min at room temperature. Blots were imaged and densitometry analysis was performed on an iBright FL1000 Gel/Cell Imager (Thermo Fisher). Data was double normalized with GAPDH as a loading control and an internal standard (IS) to control for variability between blots by combining samples.

### 4.8. Immunofluorescence Staining

Eye tissue sections were deparaffinized and rehydrated via immersion in a series of xylene, ethanol, and PBS solutions. Heat-induced epitope retrieval was performed using 1X = 6.0) in a pressure cooker. Following antigen retrieval, slides were moved into a humid chamber and washed three times with PBS. Tissue sections were then permeabilized with 0.3% Triton X-100 in PBS for 10 min followed by three washes with PBS. Sections were blocked for 30 min with a blocking buffer (PBS containing 2.5% normal goat serum (*v*/*v*)). The blocking solution was replaced with 50 µL primary antibody diluted with blocking buffer at pre-determined concentrations (Appendix A). Slides were incubated overnight at 4 °C then washed three times with PBS. Following, slides were incubated for 45 min at room temperature with a secondary antibody diluted 1:500 with blocking buffer. After three washes with PBS, nuclear staining was performed using 1 µg/mL DAPI in PBS for 10 min at room temperature. Slides were washed three times and coverslips were mounted using VECTASHIELD Vibrance Antifade Mounting Medium (Vector Laboratories, Newark, CA, USA). Fluorescent images were taken on the Olympus BX43 microscope using a 40× magnification and analyzed using ImageJ version 1.53a as described below.

### 4.9. ImageJ Quantification of Images

Photoreceptor numbers were determined in rats at the indicated post-natal ages. For consistency, slides representing the central area of the retina, based on the presence of the optic nerve as a landmark, were selected for enumeration. The 40× images were acquired roughly 1 mm superior of the optic nerve on an Olympus BX43 microscope with cellSens software version 1.18 (Olympus, Tokyo, Japan). Four contiguous sections were imaged, and the cell count was averaged for each eye measured. Retinal measurements were completed in ImageJ Fiji (ver. 2.3.0/1.53t) using a semi-automated method [60]. H&E color deconvolution is applied followed by thresholding using the Phansalkar method. Iterations of “Fill”, “Open”, and “Close” are used to create a single region of interest (ROI) for the ONL which is then adjusted manually using the brush tool. The “watershed” operation is applied to deconvoluted H&E to create a binary map of the individual photoreceptors within the ONL ROI. Finally, the photoreceptors are counted using “Analyze Particles” with a size of 1 µm minimum area. The photoreceptor count/mm is calculated by dividing the photoreceptor count by the image length (211.4 µm). 

Immunofluorescence images were analyzed in ImageJ using five images per animal. ImageJ analysis was performed by manually selecting the retinal layers. To account for the influence of autofluorescence associated with retinal degeneration, background subtraction using a 50-pixel radius. For MDA, 4HNE, and PAD4 staining, automatic thresholding using the Otsu method was applied [61]. For citH3 staining, automatic thresholding was performed using the Triangle method [62]. The percent area per retinal layer was measured and data was normalized to the mean of p21 values.

### 4.10. Statistics 

Statistical analysis was performed in GraphPad Prism 9 (Graphpad Software Inc., La Jolla, CA, USA). All graphs are plotted as mean ± standard error of the mean unless otherwise noted in the figure legend. Appropriate statistical analyses were performed for each data set. For comparing more than two groups one-way/two-way ANOVA or Kruskall-Wallis with multiple comparison corrections based on the dataset. For genetic expression data, ΔΔCT was compared using two-way ANOVA with multiple comparison corrections. The number of animals and statistical tests used in an individual analysis is indicated in the figure legends.

### 4.11. Ethical Approval

The study was carried out in compliance with the ARRIVE guidelines. Animal experiments were conducted in full accordance with the University of Southern California (USC) Institutional Animal Care and Use Committee (IACUC)-approved protocols (protocol #21045, approved 22 March 2017), National Institutes of Health Guide for the Care and Use of Laboratory Animals and the ARVO Statement for the Use of Animals in Ophthalmic and Vision Research. Dystrophic RCS rats were obtained from Dr. Matthew LaVail (University of California, San Francisco, USA) and were bred and propagated at the USC under an IACUC-approved protocol.

## Figures and Tables

**Figure 1 ijms-25-03749-f001:**
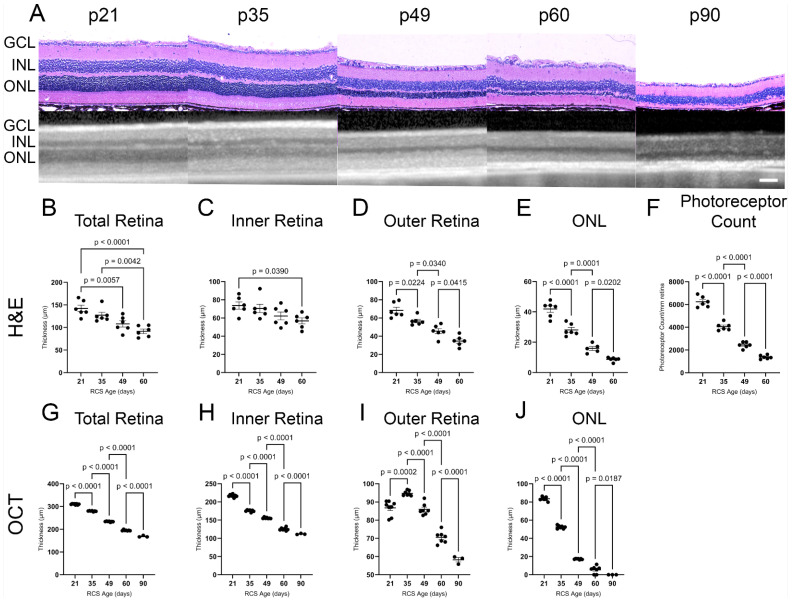
Structural analysis of RCS retina over time using histology and ocular coherence tomography (OCT). Representative H&E histology and OCT images for each age timepoint p21–p90 (**A**). Thickness of retinal layers (total, inner, outer, and outer nuclear layer (ONL)) significantly decline with RCS age as measured by H&E (**B**–**E**) and OCT (**G**–**J**) with the most drastic changes in the ONL where the photoreceptors are located. Photoreceptor nuclei counts show significant degeneration in histological analysis until p60 (**F**). Data represented as mean ± SEM, two-way ANOVA with Tukey’s correction. Histology *n* = 6 for each group. OCT imaging: *n* = 8, 7, 7, 7, 3 for p21, p35, p49, p60, and p90, respectively. Each dot in graphs (**B**–**J**) represents measurements from an individual animal. Scale bar = 100 µm.

**Figure 2 ijms-25-03749-f002:**
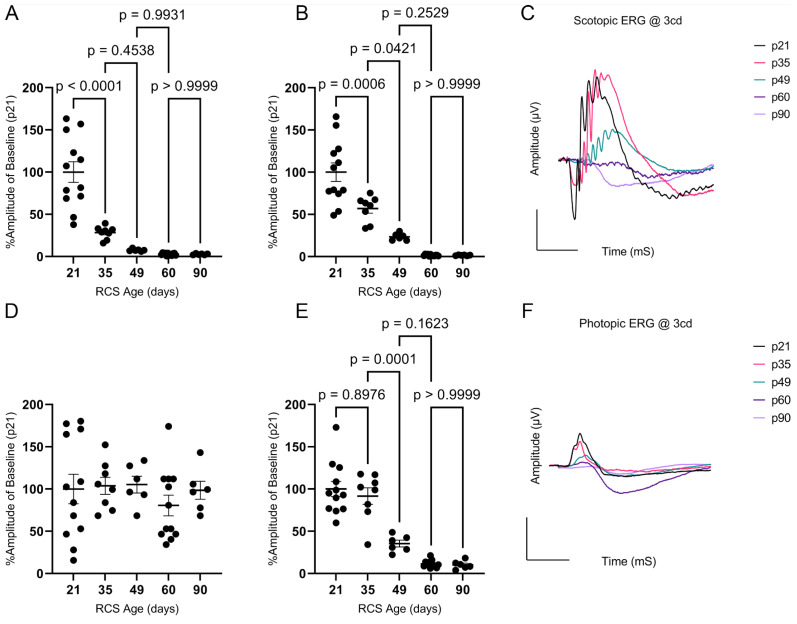
Scotopic and photopic electroretinography (ERG) analysis of RCS over time. Scotopic a-wave percent amplitudes at 3 cd∙s/m^2^ flash intensity declines dramatically between p21 and p35 (**A**), while scotopic b-wave declines similarly between p21 and p35 and p35 and p49 (**B**). Photopic a-wave amplitudes were below 5 µV at all timepoints (**D**). Photopic b-wave percent amplitudes were not significantly different at p35 but declined significantly by p49 (**E**). Representative ERGs are shown for scotopic (**C**) and photopic (**F**) waves. Data represented as mean ± SEM, two-way ANOVA with Tukey’s correction (*n* = 12, 8, 6, 12, 6 for p21, p35, p49, p60, and p90, respectively). Each dot in graphs (**A**,**B**,**D**,**E**) represents measurements from an individual animal. Scale bar = 100 µV or mS.

**Figure 3 ijms-25-03749-f003:**
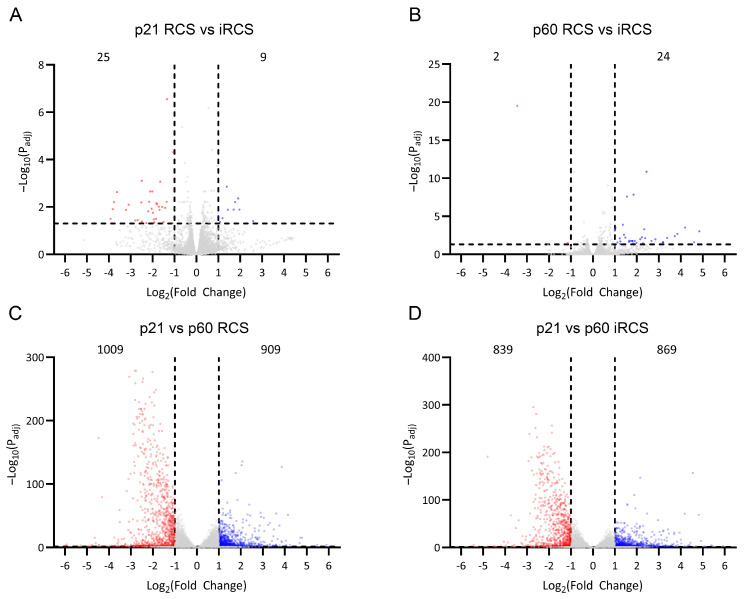
RNA-Seq differentially expressed genes (DEGs) in RCS and iRCS rats. Gene expression counts were compared between RCS and immunocompromised RCS (iRCS) at p21 (*n* = 3, each group) (**A**) and p60 (*n* = 3, each group) (**B**) which showed very few DEGs. Comparing p21 (*n* = 3) and p60 (*n* = 3) showed 1009 downregulated (red) and 909 upregulated (blue) significant DEGs for RCS (**C**) and 839 downregulated (red) and 869 (blue) upregulated for iRCS rats (**D**). Significant DEG defined as absolute fold change > 2, p_adj_ < 0.05, Bejamini-Hochberg, non-significant genes are in grey.

**Figure 4 ijms-25-03749-f004:**
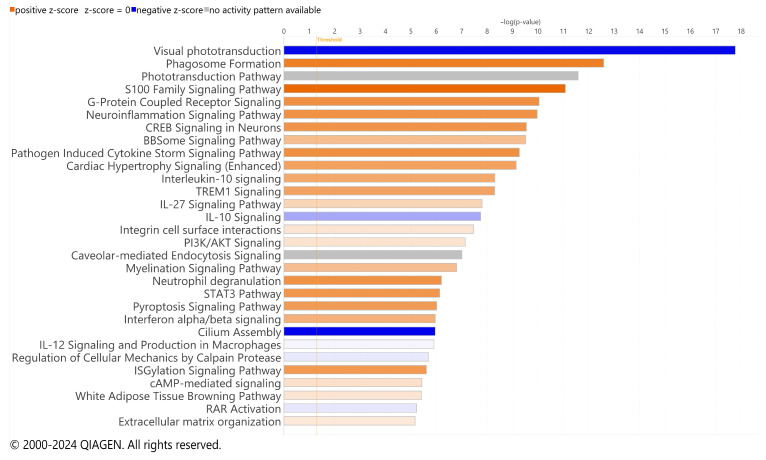
Top effected pathways in RCS rat. QIAGEN IPA canonical pathways of RNA-Seq using significant DEGs for p21 (*n* = 3) vs. p60 RCS rats (*n* = 3). Bars are colored based on z-score in which positive z-scores are orange for activated pathways, negative z-scores are blue for inhibited pathways, and when no activation pattern was determined the bars are grey. Higher intensity of the color indicates higher absolute value of the z-score.

**Figure 5 ijms-25-03749-f005:**
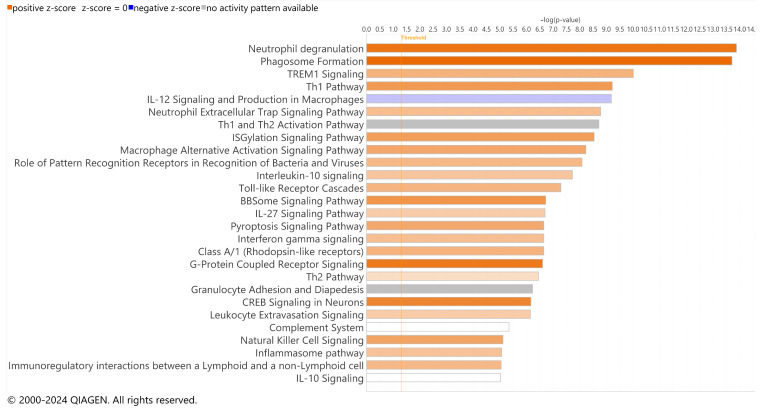
Top effected pathways in microglia-assigned cells of RCS rats. QIAGEN IPA canonical pathways of RNA-Seq using significant DEGs for p21 (*n* = 3) vs. p60 RCS rats (*n* = 3). Bars are colored based on z-score in which positive z-scores are orange for activated pathways, negative z-scores are blue for inhibited pathways, and when no activation pattern was determined the bars are grey. Higher intensity of the color indicates higher absolute value of the z-score.

**Figure 6 ijms-25-03749-f006:**
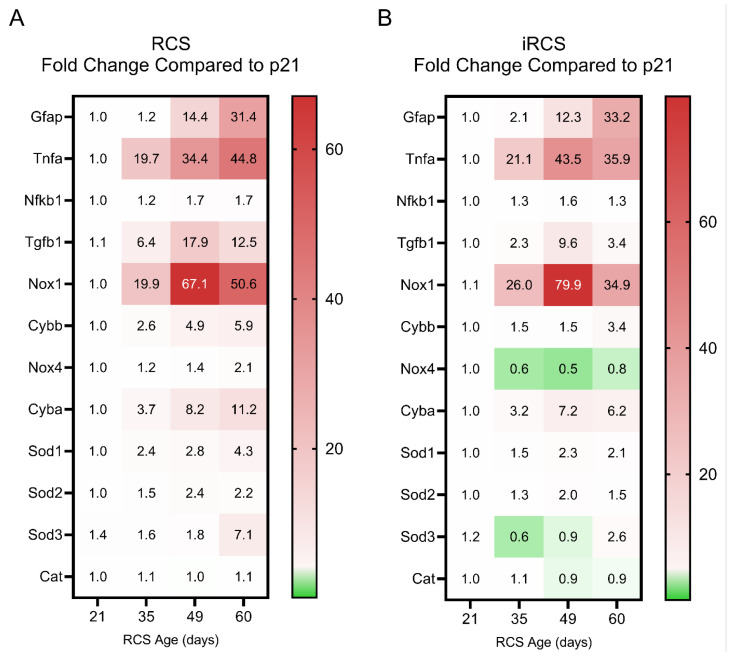
Fold change in reactive gliosis, inflammation and oxidative-stress related gene expression changes compared to p21 for RCS and iRCS rats. Heatmap of all genes displayed with mean fold change for RCS (*n* = 3) (**A**) and iRCS (*n* = 3) (**B**). Scatterplots with statistics are in Appendix A.

**Figure 7 ijms-25-03749-f007:**
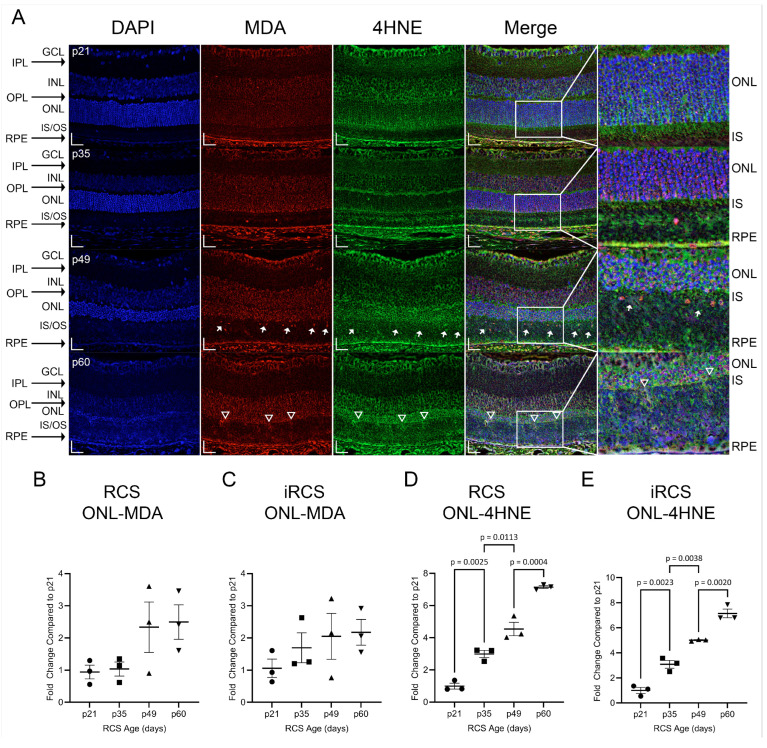
Representative immunofluorescence staining and quantitation of 4HNE and MDA in RCS and iRCS retinas throughout retinal degeneration. Immunofluorescence at each time point with DAPI (blue), MDA (red), 4HNE (green), and merged channels (**A**). MDA and 4HNE stain the GCL, RPE/choroid complex, and photoreceptor inner segments (IS) at p21. Multiple MDA^+^ nuclei are seen in the inner/outer segments (IS/OS) at p49 (white arrow) and increased staining of the external limiting membrane are observed at p60 (white triangles). Quantification of % area of the ONL, normalized to p21, did not show increased MDA staining of the ONL (**B**,**C**), but 4HNE staining of the ONL was significantly increased at each time point (**D**,**E**). Data represented as mean ± SEM, one-way ANOVA with Tukey’s correction, *n* = 3 each age group p21 (circle), p35 (squares), p49 (triangles), and p60 (inverted triangles) (**B**–**E**). Scale bars = 20 µm.

**Figure 8 ijms-25-03749-f008:**
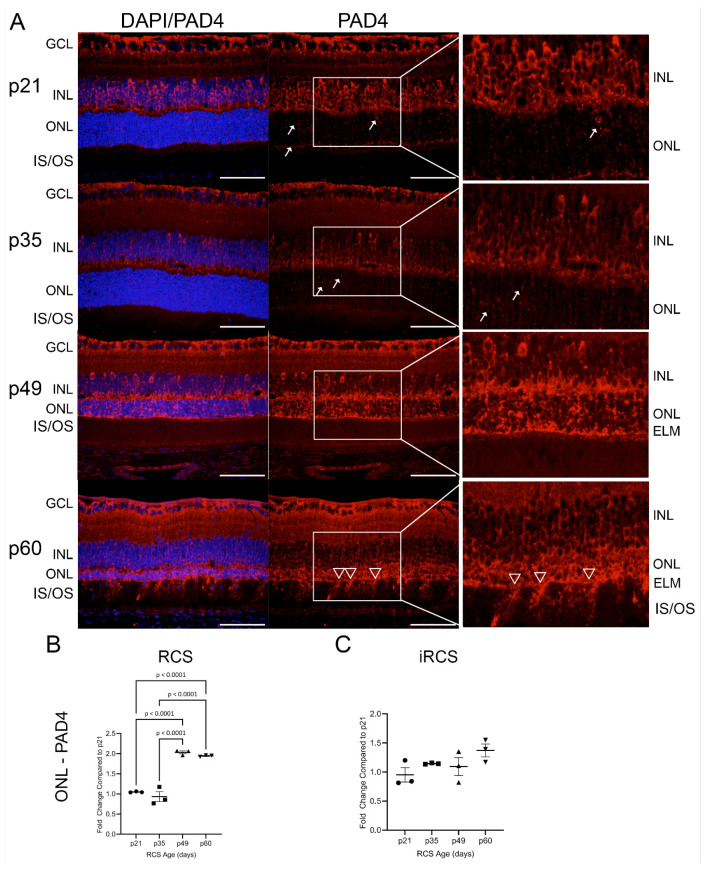
Representative immunofluorescence staining of PAD4 in RCS retinas from p21 to p60; 40× images of DAPI (blue) and PAD4 (red) (**A**). Early at p21, PAD4 is retained in the ganglion cell layer (GCL) and inner nuclear layer (INL) but focal staining in the ONL is observed at p21 and p35 (white arrows); by p49 strong expression is seen in outer nuclear layer (ONL) up to external limiting membrane (ELM), and at p60 there are regions of the ELM where PAD4 extends into the inner and outer segments (IS/OS, white triangles). ImageJ analysis show significant ONL increase in PAD4 expression for RCS rats (**B**), but not for iRCS rats (**C**). Data represented as mean ± SEM, one-way ANOVA with Tukey’s correction, *n* = 3 each age group p21 (circle), p35 (squares), p49 (triangles), and p60 (inverted triangles) (**B**,**C**). Scale bar = 50 µm.

**Figure 9 ijms-25-03749-f009:**
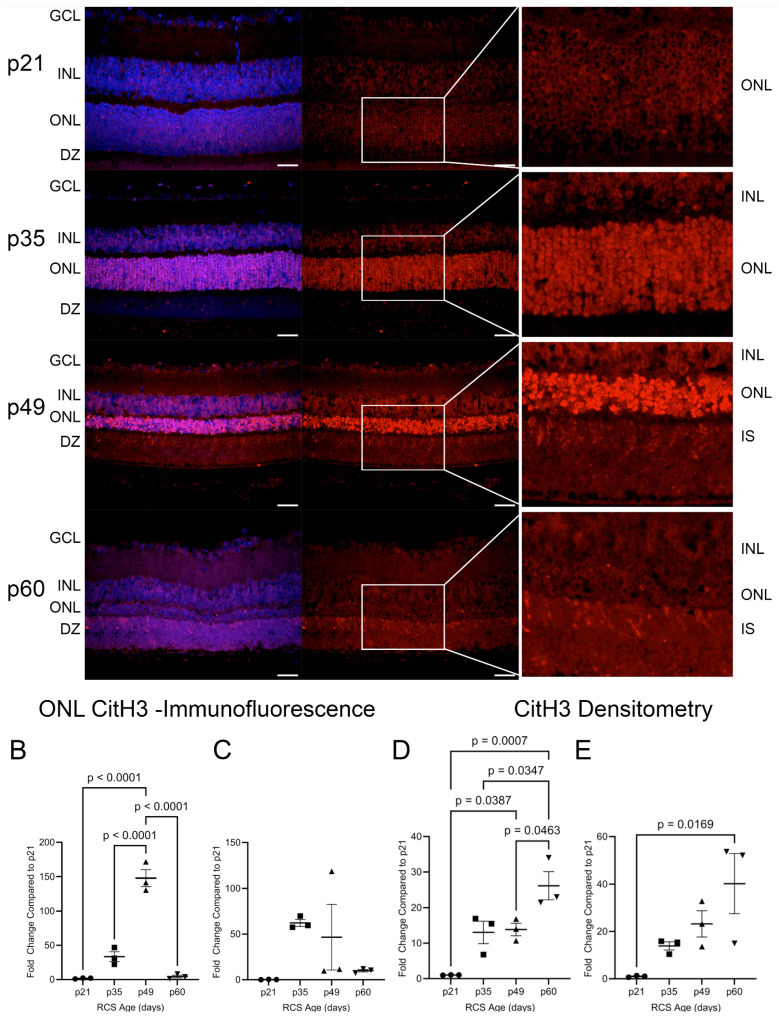
Citrullinated Histone H3 (CitH3) in RCS retinas throughout retinal degeneration. Immunofluorescence staining of retinas showed increased ONL staining at p35 and p49 (**A**–**C**). At p49 and p60, CitH3 also strongly stains in the inner and outer segments (IS/OS) and localizes to the photoreceptor IS. At p21, CitH3 was not detected by densitometry but showed an accumulation of CitH3 from p35 to p60 after normalization to GAPDH (**D**,**E**). Data are presented as mean ± SEM, one-way ANOVA with Tukey’s correction, *n* = 3 for each age group p21 (circle), p35 (squares), p49 (triangles), and p60 (inverted triangles) (**B**–**E**). Scale bar = 20 µm.

**Table 1 ijms-25-03749-t001:** Cell type labeling of RCS genes.

Cell Type	All Labeled Genes		Significantly Upregulated		Significantly Downregulated	
	Count	% of Total	Count	% of Total	Count	% of Total
RGCs	466	7%	17	4%	3	1%
Amacrine	667	10%	27	6%	17	4%
Bipolar	1462	22%	27	6%	45	11%
Horizontal	167	3%	9	2%	8	2%
Photoreceptors	26	0%	0	0%	10	2%
Cones	85	1%	6	1%	8	2%
Rods	623	9%	11	2%	217	53%
Microglia	818	12%	135	31%	29	7%
Macroglia	1811	27%	162	37%	59	14%
RBCs	7	0%	0	0%	1	0%
Vascular	534	8%	47	11%	16	4%
Total	6666	100%	441	100%	413	100%

**Table 2 ijms-25-03749-t002:** Cell type labeling of iRCS genes.

Cell Type	All Labeled Genes		Significantly Upregulated		Significantly Downregulated	
	Count	% of Total	Count	% of Total	Count	% of Total
RGCs	466	7%	12	3%	2	1%
Amacrine	667	10%	15	4%	14	4%
Bipolar	1462	22%	16	4%	38	10%
Horizontal	167	3%	6	1%	6	2%
Photoreceptors	26	0%	1	0%	9	2%
Cones	85	1%	2	0%	8	2%
Rods	623	9%	12	3%	210	56%
Microglia	818	12%	191	45%	24	6%
Macroglia	1811	27%	119	28%	53	14%
RBCs	7	0%	0	0%	0	0%
Vascular	534	8%	50	12%	12	3%
Total	6666	100%	424	100%	376	100%

**Table 3 ijms-25-03749-t003:** RCS microglial top upregulated and downregulated DEGs.

Top Upregulated Genes			Top Downregulated Genes		
Gene	Log2 (Fold Change)	Adjusted *p*-Value	Gene	Log2 (Fold Change)	Adjusted *p*-Value
Clec5a	4.7	1.02 × 10^−7^	Hk3	−1.95	1.33 × 10^−2^
Hla-dra	4.1	1.78 × 10^−11^	Ankrd2	−2.05	3.75 × 10^−4^
Icam1	3.7	2.25 × 10^−35^	G0s2	−2.14	2.31 × 10^−5^
Csf2rb	3.4	1.04 × 10^−10^	Capn3	−2.28	5.05 × 10^−34^
Mx2	2.9	3.60 × 10^−9^	Slc31a2	−2.31	8.16 × 10^−143^
Clec7a	2.7	1.41 × 10^−3^	Ciita	−2.47	1.66 × 10^−43^
Itgam	2.6	4.17 × 10^−26^	Kif2c	−2.70	7.03 × 10^−75^
Fcgr2b	2.5	1.69 × 10^−20^	C11orf98	−2.89	7.85 × 10^−18^
Galnt6	2.5	1.27 × 10^−5^	Alox15	−4.65	6.39 × 10^−16^
Cd180	2.4	2.30 × 10^−14^	Ccl5	−4.86	1.40 × 10^−3^

**Table 4 ijms-25-03749-t004:** iRCS microglial top upregulated and downregulated DEGs.

Top Upregulated Genes			Top Downregulated Genes		
Gene	Log2 (Fold Change)	Adjusted *p*-Value	Gene	Log2 (Fold Change)	Adjusted *p*-Value
Clec7a	4.34	8.56 × 10^−5^	Slc31a2	−1.94	3.00 × 10^−107^
Ccl5	3.76	9.84 × 10^−3^	G0s2	−1.99	7.21 × 10^−5^
Hla-dqa1	3.63	3.13 × 10^−6^	Ankrd2	−2.37	1.31 × 10^−4^
Cd74	3.58	4.79 × 10^−7^	Capn3	−2.41	2.25 × 10^−38^
Rgs1	3.39	1.50 × 10^−4^	Kif2c	−2.52	2.79 × 10^−67^
Myo1g	3.32	2.46 × 10^−5^	C11orf98	−2.93	1.43 × 10^−182^
Itgb2	3.03	3.39 × 10^−33^	Fam111a	−3.20	1.40 × 10^−24^
Cd180	3.00	5.73 × 10^−18^	Alox15	−3.87	4.58 × 10^−11^
Csf2rb	2.98	1.51 × 10^−8^	Cryba4	−3.91	7.42 × 10^−2^
Clec5a	2.89	1.86 × 10^−6^	Crybb1	−4.12	3.98 × 10^−2^

## Data Availability

The transcriptomics data discussed in this publication have been deposited in NCBI’s Gene Expression Omnibus and are accessible through GEO Series accession number GSE237804 (https://www.ncbi.nlm.nih.gov/geo/query/acc.cgi?acc=GSE237804, accessed on 2 January 2024). The datasets used and/or analyzed during the current study available from the corresponding author on reasonable request.

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
