# Peer review of "Unveiling Drivers of Retinal Degeneration in RCS Rats: Functional, Morphological, and Molecular Insights"

_ijms, 2024, doi:10.3390/ijms25073749_

Round 1
Reviewer 1 Report
Comments and Suggestions for Authors
The manuscript by Ahluwalia et al. investigates cellular and molecular events that occur in an experimental model of retinal dystrophy (i.e., RCS rats) while making emphasis on relevant immunodeficiency aspects. Apart from employing Transgenics (iRCS strain), the authors use a wide variety of experimental techniques. The transcriptomic analysis is timely and very much interesting, as the RCS model has long been studied using mainly electrophysiological, morphometric and imaging approaches. Unfortunately, the authors draw theoretical correlations between different findings but the experimental approach used is descriptive in nature. Why were PCA analysis done only when comparing the transcriptome between RCS and iRCS animals and not while correlating diverse features within the same RCS strain at the same age before and after degeneration? Is there a correlation between: (1) DEGs and ERGs amplitude? (2) DEGs and retinal damage? (3) DEGs and Histones Citrullination? These issues should have been addressed in the same animals in which the diverse features were assessed. The In addition to the CA analysis, the Pearson’s correlation method could also have been used. Although the data is potentially interesting, the manuscript is not well written. The text is cryptic, excessively long and difficult to understand sometimes. Some ideas appear disjointed. Most of all, some of the conclusions are not supported by the experimental data. Critical works in the reference list are missing. The reviewer cannot but advise rejection of this work for publication in IJMS. However, I provide suggestions that I hope can be useful for the authors to revise the script in future submissions.
1. – The author should down-tone the message that their findings offer valuable insights into the molecular events that drive retinal degeneration (first paragraph of the discussion). Lacking correlation analyses by PCA or Pearson’s Coefficient, the molecular events investigated in a murine model of retinal dystrophy are not so much relevant.
2. – Making a comparison between genes expressed in photoreceptors in the outer nuclear layer that eventually degenerate, and other cell types that remain in the inner retina after the degeneration process does not seem much insightful. It might be worth it to make emphasis also on what is going on at the level of gene expression in surviving cells. In the reviewer’s mind, being a descriptive study that may apply only in animals that do not even have a fovea, it lacks of general validity and interest. Any claim involving the extrapolation of animal findings into human application must be handled with caution.
4. - Behavioural analysis and visually evoked potentials are needed to support the claims of the authors. In the absence of such data, potential limitations of the study should have been discussed in the text.
5. - The authors should state clearly what strain of RCS animals were used in this study (e.g., pink-eye RCS-rdy+ strain or black-eye RCS-rdy+; see: Invest Ophthalmol Vis Sci 1981. 20(5): 671-675).
Comments on the Quality of English LanguageThe text needs extensive revision
Reviewer 2 Report
Comments and Suggestions for Authors
This study conducted a comprehensive analysis of retinal degeneration in RCS rats, including an immunodeficient RCS (iRCS) sub-strain. The analysis utilized ocular coherence tomography, electroretinography, histology, and molecular dissection using transcriptomics and immunofluorescence techniques. These findings provide valuable insights into the relationship between functional and molecular changes associated with retinal degeneration in RCS rats. Moreover, this study suggests potential therapeutic targets within inflammatory and oxidative stress pathways that could be explored for the development of treatments for retinal degenerative diseases.
Major comments:
1. The study did not observe significant differences in the progression of retinal degeneration between the immunocompetent and immunocompromised sub-strains of the RCS rat, indicating a limited contribution of adaptive immune responses to the disease. Microglial cells, which are immune cells within the central nervous system involved in inflammation and immune responses, were found to exhibit upregulation of several genes during retinal degeneration in RCS rats, suggesting their active involvement in the inflammatory response during this condition. On the one hand, there is a possibility that the sample size may not be sufficiently large to discern these differences. On the other hand, the investigation primarily relied on ocular coherence tomography, electroretinography, histology, transcriptomics, and immunofluorescence for analysis, potentially overlooking other crucial detection methods or molecular characterizations.
2. The molecular investigations in the study were limited to analyzing gene expression in inflammatory signaling pathways and oxidative stress pathways, which may have been insufficient. Further comprehensive experiments are warranted in future research.
Minor comments:
1. Line119-124, there are Errors! Reference source not found.
2. There are two figure 4 in the paper.
3. Figure 7, in Western blot analysis, the author should show the target and internal control protein expression (e.g. β-actin, GAPDH etc.) to normalize for protein loading. There are significant differences in expression levels among the same targets.
Reviewer 3 Report
Comments and Suggestions for Authors
In this manuscript by Ahluwalia et al, the authors performed analyses of the Mertk knock-out rat (RCS) with an emphasis on the structural and molecular changes that the loss of this protein evokes on the retina. While this work does provide some insight into the retinal degeneration mechanisms associated with Mertk loss, the data is largely observational in nature with insufficient rigor to support their conclusions.
To start with, as the Authors mention, Mertk is primarily expressed in the RPE and not neural retina, and it is assumed that the primary cause of photoreceptor degeneration stems from the impaired RPE function, precisely loss of the outer segment phagocytosis capability. Thus, I find the concept of only monitoring the changes in the neural retina not sufficient to fully understand the mechanisms by which the photoreceptor degeneration occurs. I would like to see similar analysis performed on the RPE cells, or if published before better incorporated and discussed alongside the authors results.
A significant portion of the manuscript analyzes the bulk RNA-seq data obtained for neural retinas at different stages of the degeneration progression (no degeneration P21, severe degeneration P60). Here, the authors focus on associating the expression level of particular gene with the major cell type associated with its expression in the retina, primarily photoreceptors. This cell-specific data is further analyzed to identify DEGs and pathway analysis. There is, however, a major flaw in this approach. The photoreceptor number severely declines (>75% loss based on Fig. 1) at P60 compared to P21, yet the decline of inner retina is much less pronounced. This means that the proportion of the photoreceptor cells to all retinal cells is much lower at P60 compared to P21. This creates an artifact in the bulk seq data for all photoreceptor-specific genes. They will seem downregulated even if their expression level in the photoreceptor does not change at P60 compared to P21. As authors rightfully mention single cell RNAseq would be a preferred approach here. In my view for the purpose of the analysis that the authors aim to perform, that is to look at the changes specifically in the photoreceptor cells, this is the only valid approach. Single cell RNA seq is widely accessible these days. In consequence, in my view, RNA seq analysis presented in the manuscript does not accurately reflect the biology of the degenerating retina.
Other comments and questions:
Line 75-76. It is unclear to me what authors mean by stating that “the sub-strains were combined for our study and will be referred to as a singular RCS strain throughout the paper, unless specified otherwise.” If this means that the two sub-strains were at some point analyzed as a single group in comparisons, this would not be acceptable to me, even considering no differences in morphological analysis.
Line 104-105 and Fig. 1. It would be beneficial to add WT (at least 1 timepoint, like P21) to some analyses presented in Fig. 1 to convince the reader to the statement that there is no degeneration at P21.
Line 108-110. This is not an accurate statement. Several (not many indeed) significant differences were in fact observed.
Line 119 and many other afterwards. The manuscript I was provided for review shows Error! Reference source not found message.
Line 124. As explained earlier these results indeed are not accurate in my view.
Line 138-141. Again, as explained earlier, due to significant loss of photoreceptors it is hard to distinguish downregulation from the unchanged expression. One could speculate whether photoreceptors do not actually overexpress some components of the pathway to compensate for the cell loss, something that would be missed in the bulk seq-based analysis.
Line 142. Please define z-score, it is not a commonly used parameter.
Line 154-155. Please provide reference and discuss more if possible.
Line 194. The referenced figure should be Fig. 5, however there is numbering mistake in figures from this point onwards.
Line 225-226. Please explain how the breaks were identified, it is not obvious to me when looking at the referenced figure. To confirm this, additional staining, for example for tight junction components, should be performed.
Figure 6. Please define what is debris zone (DZ), especially in the P21 context. Why not to call it OS/IS?
Line 522. How was RNA isolated?
Reviewer 4 Report
Comments and Suggestions for Authors
In this manuscript, Ahluwalia and colleagues show morphological and molecular insights on retinal degeneration using RCS spontaneous disease model. After a throughout investigation of morphology, differentially expressed genes and pathway analysis, the authors have found that retinal citrullination is associated with retinal degeneration. The manuscript is overall very well written and the author’s rationale to pursuit specific approaches shown, is very well appreciated. With that, I have only a few points to be considered as follows:
1. Several typos were found throughout the text, for example in lines #119, 120, 121 and 242.
2. In Figure 5 A-B the authors show 4NHE staining in p21 to p60 retinas of both groups. It is not clear if the secondary used was within the green spectrum or if the color was digitally edited after acquisition. In the case of using a green secondary, as the green channel usually overlaps with tissue autofluorescence, the authors could perhaps consider an additional western blot showing 4NHE expression in retinal tissue.
3. The use of RCS rats in two different sub strains is justified by several tests comparing both groups provided in the Supplemental figure 1. However, I wonder if the authors compared all the analysis done (especially DEG analysis) in a control rat strain as normal aging also seems to compromise the retina (Cloup et al., 2021).
4. Immunofluorescences in Figure 6 and Figure 7 show respectively increased ONL-PAD4 and Citrunillated Histone 3 in RCS retinas. Hollingsworth et al. (2018) show peak of PAD4 and overall citrunillation levels in WT around 1 month and reduction through 9-months of age. It is possible that in RCS those levels would not physiologically decline. Have the authors assessed H3Cit in older RCS mice?
Comments on the Quality of English Language
English quality is satisfactory
Round 2
Reviewer 1 Report
Comments and Suggestions for Authors
T
Comments on the Quality of English LanguageDo as you please
Reviewer 2 Report
Comments and Suggestions for Authors
The authors have appropriately addressed most of the comments and this revised paper has been improved over its original version. I have no more major concerns and feel it is now acceptable for publication.
Reviewer 3 Report
Comments and Suggestions for Authors
Ahluwalia et al describe Drivers of Retinal Degeneration (rd) in RCS Rats deficient in MERTK expression. Below are the comments to the Authors response to my initial review:
The Authors implicate they mainly focused on the fate of microglia rather than photoreceptors upon rd caused by MERTK loss, as it is central to the objectives of their paper. First of all, there is a discrepancy in the text, claiming at one point (lines 52-53) that microglial phagocytosis is affected by MERTK loss, while later (lines 68-69) that its in fact unclear. This makes the reason for focusing on the microglia unclear. Secondly, the Authors state that the objective of the paper, i.e. their analysis, is to ultimately contribute to the development and evaluation of novel treatment strategies for rd. How this would be achieved considering no clear link between MERTK deficiency-led rd and microglia? What is the proof or concept behind the idea that targeting microglia would aid in promoting photoreceptor health in such case? This rd is driven by photoreceptor loss, as shown by the Authors and others, so focus on changes in photoreceptors would be of primary importance in my view. Approach taken by the Authors requires more extensive justification.
Regarding my previous comment to line 124, I have no doubt that the Authors correctly assigned the DEGs to specific cell types, it is the usefulness of this data that is causing doubt to me. Please ensure, and assure the reader, that you actually assigned those DEGs correctly to specific cell types.
Regarding response to comment to line 138-141: The authors could correlate, or even validate observations about the phototransduction pathway, by re-analyzing their ERG data looking specifically at the leading edge of the a-wave in a fashion described by Hood and Birch (doi 10.1007/BF02584080), to look at the phototransduction kinetics.
